# The Fibromyalgia Decomposition Phenomenon: A Reflexive Thematic Analysis

**DOI:** 10.3390/bs14010047

**Published:** 2024-01-11

**Authors:** Bethany C. Fitzmaurice, Rebecca L. Grenfell, Nicola R. Heneghan, Asius T. A. Rayen, Andrew A. Soundy

**Affiliations:** 1Department of Pain Management, Sandwell and West Birmingham NHS Trust, Birmingham B18 7QH, UK; arasu.rayen@nhs.net; 2School of Sport, Exercise and Rehabilitation Sciences, University of Birmingham, Birmingham B15 2TT, UK; n.heneghan@bham.ac.uk (N.R.H.); a.a.soundy@bham.ac.uk (A.A.S.); 3Clinical Research Facility, Sandwell and West Birmingham NHS Trust, Birmingham B71 4HJ, UK; rebecca.grenfell1@nhs.net

**Keywords:** fibromyalgia, qualitative, psychology, experiences, perceptions

## Abstract

Research is needed that can provide an illustration of the different biopsychosocial and environmental experiences of people with fibromyalgia to consider how healthcare professionals can best engage with the challenges that are faced. Qualitative research is well-positioned to do this. The current study used interpretive hermeneutic phenomenology situated within a pragmatic worldview, the aim being to obtain a deeper exploration of the fibromyalgia experience prior to commencing a novel intervention. A purposive sample of individuals with fibromyalgia were selected to undertake a single interview. The interviews were analysed using a thematic analysis. The themes identified key processes of the experience. A total of 16 participants (mean age: 47.1 years) took part. Three themes and 15 sub-themes were identified, together with a process linking different experiences together. The research from this small cohort provides a clear identification of multiple components influencing the experience of fibromyalgia and the decisions around lifestyle and choices made. From this, a novel decomposition/recomposition spiral has been identified, which will benefit patients and healthcare professionals alike. An earlier diagnosis and, thus, earlier and broader treatment options can help to improve functional outcomes.

## 1. Introduction

Fibromyalgia (FM) is a broadly defined multisystem disorder encompassing persistent and widespread pain, associated with sleep disturbance, intrusive fatigue, impaired physical and cognitive functioning, and psychological distress [1,2,3]. A key development in the World Health Organization’s most recent revision of the International Classification of Diseases (ICD-11) is recognition of FM as a chronic primary pain condition. Here, chronic pain is not only classified as a disease in its own right, but equal impetus is placed upon psychosocial factors and functioning [4]. Furthermore, after 40 years, the International Association for the Study of Pain (IASP) updated their pain definition to place less onus on physical damage as being the cause of pain [5], and as such being more inclusive and expansive towards syndromes such as FM.

FM has a global prevalence of 2% [2]; this figure varies between populations and can be as high as 15% [6]. A recent cost-of-illness systematic review estimates that annual direct costs alone can be as much as USD 35,920 in the USA and USD 8504 in Europe: between two- and sixfold higher than age- and gender-matched non-FM controls globally, and higher than other chronic conditions such as cardiovascular disease and cancer [7]. The impacts of FM on individuals from a biopsychosocial perspective are considerable, and symptoms are known to be strongly influenced by culture, context, and social forces [3]. A reduced quality of life has been shown to correlate with a multitude of psychosocial variables, the most prevalent being depression, general health perception, work status, vitality, and cognitive difficulties. Pain intensity and frequency did not have a significant impact [8]. These factors adversely affect an individual’s ability to remain in employment. Around 30% of individuals (located within Europe, the USA, and Canada) are unable to work and receive disability benefits as a direct result of FM.

The paucity of functional assessment in clinical settings is mirrored in the literature, where it remains poorly defined and reported. As such, the prevalence of FM-associated functional impairment can only be surmised. One study does report on functional disability as represented by the Modified Health Assessment Questionnaire [9], a tool quantifying difficulties with physical tasks only [10]. The World Health Organization’s International Classification of Functioning, Disability, and Health (ICF) Brief Core Set has been validated as a tool to measure functioning in FM [11,12]. The ICF model encompasses functioning and disability (body functions and structures; activities and participation), and contextual factors (environmental and personal factors) [13], thus bearing the potential to classify FM functioning beyond the physical sense. A recent qualitative study assessing factors promoting physical activity showed promise towards this multidimensional approach to tailored assessment and devising patient-centred treatment, reporting on factors from both an environmental viewpoint and a personal viewpoint [14]. Early data recognise that the ICF framework can be used towards quantifying functional impairment in chronic widespread pain and FM [8,11,15,16]. However, to our knowledge, no previous qualitative studies have utilised the ICF core sets to proportionally describe the impact of FM, and then subsequently to measure a response to a therapy.

The experiences of individuals are complex, owing to the multifactorial aetiology and symptomatology, translating into difficulty in both diagnosis and treatment—each with a subsequent knock-on effect on healthcare costs [7]. Stigmatisation experiences in FM are persistently described, and the patients’ treatment journey is often complicated by this [17,18,19,20,21]. Evidence suggests that diagnostic delays lead to worse treatment outcomes [22]. The treatment experience of individuals usually involves multiple attempts at reducing pain by seeking an array of varying treatments. Medications account for a substantial proportion of medical costs [7], yet treatment-related satisfaction is poor [23,24,25,26] and high-quality evidence lacking—as described in the most recent European Alliance of Associations for Rheumatology (EULAR) international guidelines [27]. This is, in part, a contributory factor towards high medical costs—patients express low satisfaction towards treatments and continue to seek alternatives, an ineffective use of healthcare resources [7]. Delayed diagnosis, and therefore treatment, is a common occurrence [28] and leads to a more developed and severe form of FM with missed opportunities and worse treatment outcomes [22,28]. Yet, clinicians remain apprehensive to take the leap into diagnostic territory for a multitude of reasons, one being the common misconception that the diagnosis needs to come from a medical subspecialist [2]. The Royal College of Physicians’ recent guidance [2] advocates for early diagnosis and recognises the therapeutic value of this diagnostic consultation. Part of the disease-management difficulties lie in that it becomes extremely difficult to pinpoint where in this biopsychosocial “web” should be targeted first in order to produce the most significant outcomes for individuals.

Qualitative data are well situated to explore this complex “web” of an FM journey. However, research thus far is limited because it has not been able to bring together a sufficient understanding of the biopsychosocial and environmental experiences of individuals as a context which could be used to guide treatment and advice. A multitude of scientific papers [29,30,31,32,33] report negative emotions to be associated with increased pain levels, and conversely, the presence of social support reducing pain levels [30]. Research suggests that treatment progress is hindered by the lack of clear understanding of FM pathogenesis. One recently proposed preliminary psychosocial model towards FM development is the “Fibromyalgia: Imbalance of Threat and Soothing Systems” (FITSS); describing an imbalance of emotional regulation rendering the individual in constant alert mode. The authors acknowledge this is a working hypothesis, but with the potential to stimulate novel therapeutic interventions [34]. There is a further multitude of studies examining two or more FM “variables”: sleep and FM [35], cognition and FM [36], mental health and FM [37], and fatigue and FM [38]. The most recent qualitative work identifies inter-relationships between the complex physical, psychological, and social needs of FM patients and the need to further explore explanatory models [39]; the immense disruption to lives of those with FM and the inadequacy of many treatments [23]; the need to recognise comorbid FM [40]; the need for professionals to understand preferences and to receive tailored guidance towards physical activity from professionals with adequate knowledge [14,41]; FM-associated fatigue as “unpredictable”, “uncontrollable”, “unseen”, and “unintelligible” [42]; peer social support networks having positive biopsychosocial effects and empowering women to better manage their disease [43]; and the presence of a negative cycle relating to the quality and satisfaction of primary care consultations for FM, from both patient and clinician perspectives [24]. Furthermore, a recent large United Kingdom national study incorporating qualitative thematic analysis highlighted FM patients to be treated as a “low priority”, with a sizable mismatch between the needs of these patients and National Healthcare Service (NHS) provisions, most notably, multidisciplinary holistic care and community-based access to self-management. There is an urgent need to access both specialist and holistic services [44]. It is possible that by examining common experiences of individuals’ FM treatment journeys, a broader process could be identified which highlights critical aspects of the experience, which could further the understanding of treatment. It is vital that clinicians and researchers involved in FM understand the biopsychosocial and environmental experiences of individuals in order to address the current issues of poor consultation experience, delayed diagnoses, and barriers and facilitators—not solely to current treatments, which can lack efficacy, but to new and holistic treatments and to aid successful service development—in a time where the NHS is undergoing considerable reconfiguration [45] and NHS England holds patient voice in high regard as being integral towards informing healthcare services [46]. The FM voice is under-represented.

It has long been known, and more recently corroborated, that FM consultations are sub-optimal and unfulfilling for both parties for a multitude of reasons, including specialists self-professing to a general lack of understanding about the condition [44]. There remains an urgent need for healthcare practitioners to not only update their knowledge and skills to meet the currently recommended diagnostic criteria and treatment pathways, but to expand this into improved and more fulfilling patient encounters. Despite there being progress in international guidelines recognising the multidimensional experience of pain [4,26], specifically chronic widespread pain, it seems there is a lag with updated definitions failing to yet influence clinical practice. There appears to be a deficit of studies assessing the entire spectrum of FM symptoms concurrently. Further research to understand the whole biopsychosocial picture is needed if we are to target the treatment of the individual as a whole, which is a proven [8] unmet and anecdotal need for this patient group. Additionally, there is a gap in the literature between the two aforementioned identified problems [8]: poor consultations and a lack of treatment pathways. We need to explore this negative, cycling “web” on a deeper level and to identify how this knowledge can be utilised to not only interpret but to reverse this cycle.

This study aims to understand FM patients’ journeys and experiences from physical, psychological, social, and financial viewpoints, and identify processes which influence these. Further to this, this work will act as a reference point for a group of individuals who experience a relatively new therapy so that their journey can be explored with regard to perceived benefits and change.

## 2. Materials and Methods

The Standards for Reporting Qualitative Research (SRQR) [47] were used to report this section.

### 2.1. Qualitative Approach and Research Paradigm

This is an interpretive, hermeneutic, phenomenological study situated within a pragmatic world view. This world view rejects the traditional views of reality and looks to obtain practical solutions to real world problems. A qualitative approach was used in order to best explore participant’s experiences, perceptions, and opinions [48] regarding both the daily impact of their chronic pain condition and therapies trialled.

### 2.2. Researcher Characteristics and Reflexivity

Researcher 1 (B.C.F) has a background in anaesthetics and chronic pain management. B.C.F led the data collection phase, performing all of the pre-intervention interviews. Researcher 2 (A.A.S) is a qualitative research methodologist, who had no direct contact with the trial participants. Researcher 3 (R.L.G) is a clinical research practitioner who assisted with data collection. Researcher 4 (N.R.H) is a senior researcher in musculoskeletal rehabilitation sciences, with a clinical background as a practicing physiotherapist. Researcher 5 (A.T.A.R) has a background in anaesthetics and chronic pain, practicing as a consultant in chronic pain management for over twenty years to date. We decided at the outset that there would be more than one data reviewer in order to maximally utilise our broad range of strengths. We mention the specifics of where each strength added value to our analysis in the Section 2.9.

### 2.3. Context

All audio-recorded, semi-structed interviews were one-to-one and in-person, taking place at Sandwell General Hospital’s Clinical Research Facility. The interviews were taken during the first visit to a clinical setting before individuals commenced the trial intervention—photobiomodulation therapy (PBMT) [49].

### 2.4. Sampling Strategy

From January to June 2022, a non-probability convenience sample were recruited from the Department of Pain Management at Sandwell and West Birmingham NHS Trust. For the purposes of being eligible to receive whole-body PBMT, prospective participants were required to satisfy all inclusion criteria: widespread chronic pain of any origin (including axial pain, polyarthralgia, myofascial pain); able to provide informed written consent; ≥18 years; able to commit time to the trial treatment schedule of 6 weeks; and a score as low or moderate risk on the COVID-19 risk stratification tool (applicable for the duration of the pandemic). The exclusion criteria included pregnancy; severe skin diseases (e.g., skin cancer, severe eczema, dermatitis, or psoriasis); body weight ≥ 136 kg; uncontrolled co-morbidities (e.g., uncontrolled diabetes defined as HbA1c > 69 mmol/mol, decompensated heart failure, major psychiatric disturbance such as acute psychosis or suicidal ideation); the use of systemic corticosteroid therapy, including oral prednisolone or corticosteroid injections within the preceding 6 months; known active malignancy; an inability to enter the NovoTHOR^®^ device or lie flat for 20 min (either due to physical reasons or other, e.g., claustrophobia); and individuals speaking a language for which an interpreter cannot be sought (Oromo, Tigranian, Amharic, Greek). The data collection process was piloted in a sample of five participants (A–E) during the first round of eight participants. Following Trial Steering Committee meetings, it was decided to continue with data collection while new themes continued to emerge. Interviews were completed with all remaining participants to ensure data saturation was reached.

### 2.5. Ethical Issues Pertaining to Human Subjects

Ethics approval was granted by the Health Research Authority (HRA), Health and Care Research Wales (HCRW) (278452), and the Leicester Central Research and Ethics Committee (21/EM/0231) (ClinicalTrials.gov trial registration number: NCT05069363; 6 October 2021). All participants gave written informed consent and were free to withdraw from study participation at any point. At each interview visit, it was re-iterated that the interview was wholly optional, and that the participant could stop the interview and recording at any point.

### 2.6. Data Collection Methods

Participants who opted to be interviewed during the consent process underwent semi-structured, audio-recorded interviews at three stages of the trial: “pre-intervention” at their first visit, “during intervention” at treatment visit 9, and “post-intervention” at their final visit. The interviews took place from 31 January 2022 through to 29 June 2022. The current study focuses on the pre-trial experiences that lead up to the individual deciding to undertake a novel therapy, revealing their journey until that point. The interview guide was developed by experienced qualitative researchers based on research objectives (see Appendix A). The “pre-intervention” interview comprised three main sections: introductory questions, transition questions, and main questions. The section “main questions” encompassed three sub-sections: “pharmacological and injection therapy”, “non-pharmacological therapies”, and “about the trial and the device”. The interview schedule in its entirety included 17 questions. The first section opened with open-ended introductory questions in order to gain insight to the participants’ disease experience, followed by the transition section to hone in on daily activities that are limited, along with any coping strategies. Next, the participants were asked about pharmacological and non-pharmacological therapies, past and present, finishing with factors specifically related to the “light therapy” and motivations for entering the trial. The interview schedule remained consistent for the duration of the trial. No adaptations to the questions were made, but as the interviewer (B.C.F) progressed through participants, more onus was put onto current issues relating to living with FM, rather than too much focus on past events. Demographic data were collected both verbally and from clinical records with the participants’ consent.

### 2.7. Data Collection Instruments and Technologies

The audio recordings were via Microsoft Teams for the purposes of transcription. The primary researcher (B.C.F.) collected all data for the pre-intervention dataset. The interviews lasted an average of 38 min, ranging between 17 min and 58 min.

### 2.8. Data Processing

Data from the interviews were transcribed by the authors B.C.F. and R.L.G. The authors primarily adopted a reflexive thematic analytical approach [50,51], supplemented by guidance on quality [52] and a pragmatist approach [53], which was felt suited to both the dataset and targeted audience. The interviews and transcripts were stored on a secure password-encrypted NHS computer. For the purpose of the write-up, the participants’ trial codes were further anonymised with a separate coding system.

### 2.9. Data Analysis

We describe our six-phase analytical process [51] in Appendix A, namely “Familiarisation with the data”, “Generating initial codes”, “Generating themes”, “Reviewing potential themes”, “Defining and naming themes”, and “Producing the report”.

### 2.10. Techniques to Enhance Trustworthiness

The trial was monitored and audited by the Trust’s Research Management and Governance Facilitator pre-trial, mid-trial, and post-trial. An audit trail was documented throughout the analysis. Each amendment to the analysis was saved as a separate document in its own right so that the processes are clearly transparent.

## 3. Results

The following section is detailed secondary to the nature of the rich data obtained. The main points of focus for implications are the over-arching themes identified and the discovery of novel processes surrounding FM.

### 3.1. Description of Participants

This qualitative study included 16 participants. One participant was excluded following the first visit and treatment visit 9 interviews due to exiting the trial prior to the completion of therapy. Three participants were not interviewed due to a simultaneous and ongoing evaluation of pilot data. Table 1 summarises the demographic information of the participants.

### 3.2. Themes

We identified three over-arching themes, with 15 sub-themes (Figure 1), in accordance with ICF domains in order to capture functioning and disability associated with FM. At the time of interviewing, the domain “body structure and function” played a significant part in inhibiting the individual’s functionality; this is represented by the substantially higher proportion of associated sub-themes, pictured as a “web”. “Activities and participation” represent difficulties an individual may have in executing activities and involvement in life situations [16], respectively. Environmental factors are defined as “the physical, social and attitudinal environment in which people live and conduct their lives. These are either barriers to or facilitators of the person’s functioning” [16]. Sub-themes were then created, modified, and agreed upon by Reviewers 1–4 to represent our dataset. Here, we present sub-themes individualistically. We expand on the real-world complex interplays between themes in Figure 2.

#### 3.2.1. Body Structure and Function

1. Pain: The experience of pain was regarded as limiting the choice and nature of activity (B; I; J; K; M) and had an unpredictable impact on work duties (K; P), or even the ability to make it home (A). Participant A explained *“it even hurts just to get upstairs”.* When pain levels were perceived as “too high”, activity could cease as a way to manage the pain (A; B; I; K; M), with some participants not leaving the house, or even leaving their bed. Participants stated that headaches could prevent activity, until the pain eases off. This could be 2–3 days (B) or half a day per week in bed (D). Further to this, pain could affect certain and alternating parts of the body, and this varied in the impact it had on activity (A; B; E; F; G; I; L; M; N). An example from Participant B describes, *“headaches are quite severe, pain in my elbows, skin sore to the touch…sometimes I can’t touch my head ‘cause it’s that sore…pulsating pains down my legs”.* The experience of pain is uncertain (A; B; C; D; I; N), and the number of days affected by pain each month can, and will, vary (B; I; J; L; M; N). The pain response can be disproportionate to activity levels (G). Weather could dictate pain and, hence, activity levels (J; K). One participant identified being constantly defeated by pain, with no relief from it (I).


*“my pain’s that bad…I just think—no, I can’t be bothered”*

*(Participant A)*



*“very tight and aching as if somebody needs to roll my body out with a rolling pin…it’s kinda like you’ve got the flu—heavy and achey…random spasms”*

*(Participant D)*



*“the other pain is in my scapula and neck area where you feel like somebody using a chisel and digging down in my flesh, and other sensations—like a hot piece of iron on your flesh…my jawbone, my face is painful…I find it painful talking, eating, and if I like cough or yawn it hurts, it’s so painful”*

*(Participant H)*



*“when I wake up there’s so much pain, it’s like someone’s been stepping on me—it’s crazy, crazy pain”*

*(Participant L)*



*“the pain is just constantly there…it’s amazing that your arms hurt when you walk. Why do your arms hurt when you walk?!”*

*(Participant P)*


2. Symptoms flaring up: This sub-theme represents a high degree of inter- and intra-variability in flare-up symptoms, both temporally and in intensity. Symptoms could flare up all at once, leaving the individual incapacitated (M). A context to this is provided by Participant F, who stated, “*A really bad day looks like—I can’t stand, I can’t even wipe my own bottom, I don’t want to be here. Everything sort of kinda like shuts down. The light hurts my eyes, I can’t lift up my head, I can’t get comfortable. The pain even in my toes, it’s just everywhere*”. A flare-up means 2–3 days in bed; the flare-up normally lasts 48 h, but at worst, 4 weeks, with a frequency of every two months. With a bad flare-up, personal care is impossible (H); symptoms include constant pain (J), exhaustion and an inability to leave one’s bed, and an inability to work (K). The variation in experiences continued; for instance, Participant M identified that they happened 5–6 times a year, whereas Participant N said they were 1–2 times per week and could last an hour, and Participant I perceived the flare-up as constant, with the breaks in the flare being identified as only a few days, and exacerbated by other illness such as infections. Other varied symptoms were also considered; for instance, Participant M stated that a flare-up consisted of constipation, confusion, memory problems, joint pain, and physical fatigue. The majority of individuals adopted a passive approach to flares—*“when I get headaches it just completely wipes me out, I can’t do anything just ride through it, but sometimes it takes 2 or 3 days…I’ll just lie on sofa, wrap something around my head, painkillers…just have to wait until it eases off”, “can’t get myself motivated and up out of bed…tiredness just unbelievable to a point where I can’t function”, “I just feel as if I’m wearing a lead jacket…so I just end up basically sitting on the sofa or in bed”.*


*“Exhaustion, the pain’s a lot worse, can’t get out of bed. There’s no way I could make it to do any work even though I’ve just got to go downstairs…I’ve had to phone work and just say I can’t physically work”*

*(Participant K)*



*“Sometimes I can’t get up the stairs. Sometimes I have to come down on my bottom…I can just walk sometimes and my legs give way”*

*(Participant P)*


3. Restricted physically: This sub-theme identified how the FM symptoms, predominantly pain and stiffness, inhibited physical function and the ability to undergo usual activities (A; B; C; D; E; F; G; H; K; L; M; N; O; P). Individuals identified an inability to undertake activities of daily living (F; H; I; J; M; N; O; P)—*“I can’t carry nothing like a washing basket in my arms…shopping’s a big thing for me ‘cause I can’t carry”, “even doing the washing up I have to sit down about 4 or 5 times”*, *“carers…help me shower, wash my hair and brush it because I can’t lift me arms. They help me get dressed so like my bra and knickers”.* There is difficulty with particular interactions, such as a physical difficulty with small children (F; L; P) and pets (A; C; G; K; M). The restrictions included a physical inability to continue (E) and represented a sharp contrast from being active to being inactive (C; N; O).


*“I used to walk my labrador quite a distance across the canal—don’t do it so much now because of the pain”*

*(Participant A)*



*“I could be walking and I just get tightness from nowhere and muscle pain…I just can’t seem to identify exactly how it’s happening”*

*(Participant B)*



*“I get support with some personal assistance, like to do housework, to cook and to shop”*

*(Participant H)*



*“I can’t walk very much. I can maybe walk from my house to the corner shop…maybe 15 yards and that’s it I’ve had it”*

*(Participant O)*



*“It’s really difficult for me to wipe my own backside”*

*(Participant N)*


4. Body sensitivity: This sub-theme represents a hyperactivity of the nervous system with its resultant effects. It identifies how individuals expressed being sensitive to multiple and unique experiences. This included sensitivity to (1) smells (D; P) to a point of making the individual feel sick (D); (2) light, hurting eyes and causing headaches (D; F)—Participant F explained, “*Because the light was hurting my eyes and I couldn’t move and I actually wet myself. And I’ve never done that ever. Ever…All I know, I remember the light was hurting my eyes and I couldn’t get out of bed*”—(3) to pain, causing an over-reaction (G); (4) to touch and pain during touch (B; J; L), to cold or hot (I; L); or to itching (K; P) (as Participant K stated, “*my body is marked everywhere from where I scratch*”); (5) and to noise (K; P) and a need to have a quiet environment—interfering with the ability to cope at work. Interestingly, this final problem occurred in a participant that was previously known for the ability to multi-task “*my husband used to go mad because I used to sit there with the television on watching a programme and typing at the same time—he used to say how do you know what you are typing? You ain’t looked at the screen*”. Experiences in the body further varied and were significant, like feeling exhausted and having a heavy body (D), or having a body that became extremely hot (N). Participants commonly described restlessness and even involuntary movements. They identified a hypersensitivity of their body resulting in decisions being made to adjust routines and behaviours (B).


*“My skin literally burns like on fire. It crawls all over my body”*

*(Participant H)*



*“It hurts even for a thin light quilt to rest on my feet”*

*(Participant P)*


5. Body limits: This sub-theme represents participants identifying limitations of the body and a contrast between what their body could once do and what it can do now (C; D; E), occasionally being described as an ongoing battle with one’s own body: *“sometimes you just want your old body back…I’m just fighting with it, that’s how I feel, I’m just constantly battling”* (Participant C). Two participants identified a loss of trust in their own body (A; I). The body was identified as something that makes decisions about what can and cannot be done. This could include activities like swimming, falling down (E), or sleeping (G): *“I wanted to sleep…my body was just like—no”*. Participant C identified that they could no longer drive a car, and Participant J identified that specific clothes are too restrictive to move in. The body was identified as being out of control (G; J) or dictating limits, for instance becoming hot, shaking, or being tender (I), and showing that nothing can be done when pain takes over (E).


*“I just do what needs doing in little chunks because if I try to do too much in one go it’ll backfire on me”*

*(Participant B)*



*“I’d feel as if I’d hit a brick wall…my body’s kind of on its brink”*

*(Participant D)*



*“I can’t do anything to un-knot it”*

*(Participant E)*



*“I lose the ability to walk…my knee will just jolt out and I’ll fall down”*

*(Participant G)*



*“You know it’s like as if my body’s in control of me, not me in control of it”*

*(Participant J)*


6. Time needed to mobilise: This sub-theme represents lost productivity in activities due to the length of time (up to three hours) taken to become active in the mornings (B; E; F; I; L; M; P), along with worse “fog” in the morning (H). Individuals identified physical movement challenges and slowness of movement (E; F; I). Individuals also identified challenges like entering the bath or shower (E; F; I) or dressing themselves (E). Physical challenges were not solely responsible for this, with Participant B describing, *“it takes me ages to get in the mood to have a shower”.* The act of washing and dressing exacerbates pain, and therefore, some participants see this as a chore and neglect to do it (K; L).

7. Low mood, motivation, and energy: This sub-theme represents the identification of poor mood and motivation (A; B; C; E; F; G; H; L; M; O; P), and anhedonia for previously loved work and hobbies (B; N; P), with associated guilt for “letting people down” (F; P), and being unable to control emotions, specifically agitation (D; J ;N). Experiences in the body further varied and were significant, like feeling exhausted and having a heavy body (D), or being constantly on edge, making the body feel even more tense (D).


*“I ain’t there just yet to do it again (yoga)—I’m not motivated”*

*(Participant A)*



*“I’ve got no interest in anything…if I haven’t got a reason to get up I’ll stay in bed all day…well because of the quality of my life, I just think you wouldn’t let an animal suffer like this. I welcome death because of the pain”*

*(Participant F)*



*“it took me 3 days to clean up my bedroom and I didn’t wanna do it but I had to ‘cause it was a mess”*

*(Participant G)*



*“I get really fed up and snappy…I snap at people for nothing…agitated at the least little thing”*

*(Participant O)*



*“If you said to me like here’s a gun you can shoot yourself, I probably would—it’s not being able to cope with the pains”*

*(Participant P)*


8. Poor memory (dyscognition): This sub-theme identifies the impact of how poor memory acted to inhibit conversations, hobbies, and vocation. It was identified as influencing confidence for future interactions and occasionally feeling stigmatised. Examples of experiences included (1) not being able to remember important events, names, or important information, such as the side of the road to drive on and one’s place of work (A; B; F; H; K; L; P); (2) communicating with others, including mixing up words (A; B; I; J; K; P); and (3) not being able to remain attentive to tasks undertaken or interactions (B; D; E; K; L; M; O). Participant H assigned poor memory and concentration as the barrier to completing a degree. Finally, one participant (E) identified both medications and not being able to stop thinking about pain as exacerbating factors—*“I remember the past but couldn’t tell you dates, but I think some of that’s the morphine as well ‘cause that definitely had an impact”.* Several participants required questions repeating halfway through giving an answer—*“what was the question sorry?”, “my mind just goes blank, completely blank. It’s gone blank now”.*


*“I get mixed up with words. I don’t open up in conversation because I feel ashamed”*

*(Participant A)*



*“I just can’t stay engaged for that long, I can’t stay interested, my attention span’s gone”*

*(Participant B)*



*“My reactions would feel as if they were a lot slower…I’d feel kinda too tired to answer straight away. I’d hear somebody saying something and I’d have to think it through”*

*(Participant D)*



*“I can ask what drinks people want, walk into the next room, then I haven’t got a clue what they want…I’ll have to re-read a lot ‘cause I don’t remember the page I’m on…I forget the names of words a lot as well—my degree’s in Psychology and English—you’d think I’d know words”*

*(Participant E)*



*“Grandchildren’s names? I’d never remember…I repeat myself a lot…I’ll tell them 3 or 4 times and not realise I’ve already told them…I used to love reading, that was my favourite thing, just get lost in a book—I can’t now I couldn’t even read a page, if I read a line I forgot what it says so I go back to the beginning”, ”I think I annoy people, and my speech comes out all jumbled up…so I think I’m saying the right thing but something else is coming out and I don’t realise”*

*(Participant F)*



*“I mix up names of things but I don’t realise unless somebody points it out…like the other day I said I’m going to put my pizza in the fridge but I meant the oven—like it wouldn’t click in my head that I’d got the wrong word”*

*(Participant I)*



*“I couldn’t string sentences together and I thought have I got dementia?”*

*(Participant J)*



*“um what year are we in now? I was gonna say 1919, but it ain’t 1919 is it?*

*(Participant P)*


9. Unpredictability of fatigue and sleep disturbance: This sub-theme identifies fatigue and sleep as two dominant features towards creating the vicious cycle that impacted on wellbeing and quality of life (E; F; G; H; I; J; K; L; M; N; O; P). Some participants (E; M; O) feel that fatigue is the most predominant and intrusive FM symptom. Some identified that it could take a long time to fall asleep despite feeling exhausted (K; L; M). Sleep was further complicated by the experience of pain. Specifically, individuals identified that they simply cannot fall asleep despite their sheer level of exhaustion (B; D; M); pain and cramps disturb sleep several times per night, and sleep is light and broken (B; C; D; F; G; M), leading to exhaustion (B; D; I), and sleep is not refreshing. Participant A describes, *“I get up but I’m still as tired as I was before I went to bed”.* Some participants attribute their poor quality, unrefreshing sleep to their medications (D). Several individuals describe feeling significantly worse after a night in bed—*“I feel like I’ve been beaten up in my sleep…I feel like I’m 100 when I wake up in the morning”, “I wake up…as if I’ve gone to the gym and done too much”.* The impact of fatigue and sleep disturbance was considerable. For instance, individuals identified and recognised a close linkage and knock-on effects between symptom domains (D; L; M), over-medicating at night in order to induce sleep (J), poor sleep hygiene (K; L; N), and needing to nap during the daytime, even at the cost of leaving work early (B; J; K; M; N), or even during conversations (A) and work duties (F; K). Fatigue appears to come in unpredictable waves—*“I’ll carry on until all of a sudden I start just feeling really tired, my energy level just drops off a cliff and I need to go and lie down”.* Further to this, some individuals identified a reliance on others due to severe fatigue (C; I; K), and detriment to personal hygiene is becoming the norm (K; L)—*“some days I don’t even shower ‘cause I’m that tired…don’t bother brushing my teeth. I could go days without showering and cleaning teeth. Even when I know I’ve got to work the next day sometimes I just put on plenty of perfume, dry spray, and I feel disgusting—and I just can’t physically help it”.*


*“By lunchtime I’m fatigued so I have to go home and have a nap”*

*(Participant B)*



*“I think it was yesterday I didn’t wake up until 7 pm on the night-time. And I thought I was gonna be awake all night and I wasn’t. I fell asleep at 3 am and woke at 10 am today to get here so I’m actually shocked that I still feel exhausted. I feel like I’ve had no sleep for weeks”*

*(Participant G)*



*“Tired, no energy I just feel dead, deflated…flat”*

*(Participant H)*



*“The pain just from showering can exhaust me”*

*(Participant I)*



*“I know I shouldn’t have but I just took another two co-codamol and another two gabapentin 600 mg, I think that’s about 1200 mg…don’t feel refreshed, don’t feel as if I’ve had a good enough sleep, I just feel like the soreness of my muscles is preventing me from completely relaxing”*

*(Participant J)*



*“There have been nights where I have gone straight through without sleep”*

*(Participant K)*



*“Sometimes I’m really really tired I say to myself I’m gonna go straight in bed when I leave work, the tiredness is crazy”*

*(Participant L)*



*“I do nap in the day, my wife calls them cat naps ‘cause I’ll just be reading my phone…and the next thing I’m just asleep. There’s time when my wife says I’ve gone to sleep with my phone in my hand still pointing up at my face”*

*(Participant M)*



*“I mean some nights I don’t even go to bed, I stop downstairs ‘cause I know I ain’t gonna go to sleep so it’s a waste of time…I feel like I’ve got no energy whatsoever”*

*(Participant O)*


#### 3.2.2. Activities and Participation

1. Social loss: This sub-theme identifies a change in, and in several cases, a loss of, relationships (D; K; O; P), including spousal (B) and friendships lost from an early age (G; H; I). A shift in relationships could occur due to not being able to undertake hobbies and prior events that were integral to their identity (H; P), or the physical inability to have contact (E) or connection with others, like lifting children (L) or not being able to hold a conversation with others (A), being desperate for visitors to leave due to having to put on a brave face (H). For some, it was better not making plans, for instance, to play with children or take them to activities, which was considered to have a knock-on effect on the child’s mental health (H; I; L; M; P). Significantly reduced confidence in physical and cognitive abilities led to a loss of spontaneity (D; P), pre-emptively contriving excuses (G), social anxiety (I; K; P)—*“I go back into my shell…move away from everybody”*—events not being worth it due to after-effects (H), and having to factor in subsequent rest days: *“I plan it around when I’ll be able to have respite”* (I). This participant explained, *“sometimes I have a meltdown thinking about how much pain I’m going to be in”.* Participant F simply said, “*I don’t like people…I leave house for appointments and that’s it”.*


*“I put things off, I’ll make things up…I feel as though I’ll let people down before it’s even started…we’re supposed to be going out this Thursday and I’m dreading it even now ‘cause they’ll talk about something and my mind’ll just go”*

*(Participant A)*



*“Some days it’s like I want to go, we book a table then I get to the night and it’s just like—I can’t. Sometimes I back out of it…sometimes I wonder whether it’s safe to drive the car…I get to the point where it’s just like I can’t function”*

*(Participant B)*



*“I just wanna curl up into a ball and stay under my safety net”*

*(Participant G)*



*“Church…I went yesterday…I came home and couldn’t eat, my body hurts all over. I was saying this is what happens when I go out on a social you know—I get this impact. Best not to make plans”*

*(Participant H)*


2. Vocational loss: This sub-theme highlights how some individuals identified the loss of or change in their job. There were several reasons for this, for instance, a loss of passion and motivation for a job (B; D; P), the physical inability to do anything other than a sedentary job (F; O; P), or experiences like falling asleep on the job (F; K), having no job since a young age (E; G), and being scared of commitment to a job (E), with pain and exhaustion being worse during the job (C; K). Several participants identified feeling victimised at their workplace (A; C; F), and in some instances, this lead to their departure (C; N).


*“Eventually they had to let me go…when I was off other people had to do my work…feeling guilty that I’m letting everybody down”*

*(Participant B)*



*“Sometimes I just have to go to the toilet and lock myself in for a while and just cry. I have to force myself to stay because I know I’m on my last warning”*

*(Participant F)*



*“sometime I just do hit a wall and I just have to close my eyes even if I’m working. You know my hands like doing this [hand action] moving the mouse so they think I’m working”*

*(Participant K)*


3. Reduced activities with others: Losses were counted to include no longer being able to facilitate a child’s hobbies (H; P), which causes anguish on both parts—*“withdrawn, upset and angry”*—whilst some activities were avoided as a reliance on family could increase (I). For others, there was intimacy (M), and the desire for normal engagement in activities had disappeared (M), replaced by a desire for quietness or peace (B; F; L; N)—*“I’ve got to the point now where I wanna just store myself away…half the time I don’t wanna go or I can’t make the effort”, “if somebody comes to visit I’m not as engaged…it seems too much effort”*. For some, this could include sleeping in separate beds (F).


*“My children…they say I’m gonna do something—then I don’t do it…I should be over the park with the kids and the dogs and I can’t do that…impossible…never”*

*(Participant F)*



*“If I go out [e.g., shopping] it’s like I’ve walked up the highest mountain in the world and back in a day…I’m laid up in bed all the next day”*

*(Participant G)*



*“My son, he wants to play football in the park and I say no to him like, but he doesn’t understand—he thinks I’m lazy”*

*(Participant L)*



*“Me and my wife used to have a relatively good sex life…but not anymore, it’s just completely changed that side of the relationship…reduced libido, reduced urge, even pain…it just makes everything physically undesirable”*

*(Participant M)*


#### 3.2.3. Environment

1. Lack of understanding: For some, there was a lack of understanding from family as to why they would not engage in activities (A; B; K; L; P). For instance, this included avoiding holidays or social gatherings (L), or rearranging activities to avoid going out (M; N; O; P). Individuals identified annoying family members (F) with poor memory (K), feeling misunderstood (B; K; P), not wanting to see family (B; L; N; O), or even asking family members to help them commit suicide (F). There could be a contrast of not wanting to rely on others (G) but having to rely on others (F; G; M). Participant A identified that, due to their low mood, judgment was clouded, and in relation to her FM symptoms, sometimes thought *“I was making up a thing”.*


*“I just feel I’m alone”*

*(Participant A)*



*“Nobody can understand you know, ‘cause you look physically fine”*

*(Participant B)*



*“My son says ‘I already told you mum and you’re just not listening to me’”*

*(Participant K)*



*“Struggling with conversations…I haven’t got anything to say…I mean ‘cause they’re fed up of hearing it I think”*

*(Participant N)*


2. Not being believed: This sub-theme represents self-doubt in their own diagnosis due to the perceptions of others (A; B; C; N). Individuals expressed that others could consider that they are exaggerating their diagnosis (B), or others not believing what is happening because of the lack of external physical signs (B; N; P)—*“they ain’t bothered, they just think like ‘oh, you go to work every day so you’re OK’”*. Further to this, there were experiences of medical professionals not believing symptoms. Participant D stated, “*Every doctor I spoke to told me—‘oh it’s in your head. Take a paracetamol’…so first of all they thought I was going crazy*”. This perception led to behaviours where the individual would not let family members see or hear about their experiences (F; G; J; N; P). Participant F stated, “*I don’t let them see…they know something about it, but I try and hide. They don’t hear me crying with pain*”. Alternatively, some would lie to others in order to be able to leave a situation if they needed to (G). However, such behaviours had subsequent knock-on effect on relationships (F; J; K; L; N; P).


*“People don’t wanna hear it and they think I’m making it up…other partners aren’t supportive because they don’t believe it’s a thing. They think it’s made up because you can’t be bothered”*

*(Participant A)*


3. Strained interactions: Strained relationships were identified in the majority. Several participants felt that they were agitated against their will, even at young grandchildren. Actions and behaviours of the individual can be associated with anhedonia towards being with others (B; F; K; L; O)—*“I used to go out with my sisters…I think it’s gotta be nearly 7 years ago…that’s just who I am now, I don’t wanna be there”*. Participant A stated that people do not want to listen to you and talk about the problems. Interactions could develop where the other person needs to “push” the individual to do anything (A). Examples of this included a partner becoming frustrated because they cannot do what they used to (C); individual interactions with family could become more difficult, as Participant K explains, “*my husband can’t sleep at night because I am constantly crying in pain*”, or the individual cannot engage with grandchildren or children (F; H; L) or be with them during important moments, like national-level sporting events (P). Individuals identified that they could become very difficult to live with, yet in need of others, and there is a great concern for the impact it has on them, for instance, *“when you can’t wipe your own bottom”* (F). Individuals stated that they were less in the present, less engaged with people (B; L), including children (H; J), or had more difficult interactions; for instance, Participant D stated, *“I’m not a nasty person, but I can be very abrupt…I don’t know whether it’s ‘cause I’m constantly on edge”.* This took joy from the interaction and could leave the individual perceiving that they annoy people (F) and become sensitive to this (K). The context for this was important to consider; for instance, Participant E stated, *“even hugs hurt”.* Commitment to activities and engagements is problematic; when the time comes, it can be too much, with too much pain (A; G), pre-emptively worrying about just how much pain they will be in if the activity is undertaken (I), and with a track record of learned behaviours that with undertaking activities comes pain (H;I). Strained interactions are not only limited to family and friends, but extend to contacts with certain caregivers. Participants feel that different branches of one condition are treated as separate entities: *“I had physios at work, I had physios through the insurance company, several physio visits in the NHS as well…some of them—they’ll be of the opinion it’s not FM, it’s just muscle pain or whatever…so they don’t tackle it like how they should and it kinda makes it worse”* (Participant B).


*“He can’t cope with me saying things like ‘I just can’t live like this anymore’—so it’s difficult for me to speak to him…it would hurt him and he’d worry all the time”*

*(Participant P)*


### 3.3. The Thematic Synthesis

Figure 2 illustrates two cycles of how specific ICF domains were identified to interact in order to either restrict (cycle 1) or enable (cycle 2) the individual’s choices, interactions, and activities. The model suggests that a “decomposition reaction” occurs between the ICF domains of the body’s function and structures; activities and participation; and the environment. This is contrasted to a proposed model, a “recomposition reaction”, which was supported by far less evidence. We use this term to illustrate the process of being able to undertake activities, interactions, and choices in an adapted or less restricted way.

FM has become an integral part of the participants’ self-identification (I; O; P). There is a defeatist attitude*—“I don’t even care about the way I look now…I was always like a size 12. Now I’m a 16/18—I’ve put loads of weight on ‘cause I just ain’t bothered”*—and it seems that participants have succumbed to their diagnosis and allow the pain to dictate their activities (B; C; D; F; H; L; M; P); *“I still liked my job…but more of my concentration was on managing the pain rather than doing my actual job”*—this particular participant has three degrees related to their prior job that they had to discontinue; *“I never really have a good day because the pain is still there”*.


*“It’s absolutely changed my life because it feels like somebody pulled a rug from beneath my feet…it hasn’t allowed me to finish my degree that I wanted to finish in 2013”*

*(Participant H)*


Participants lack acceptance of their diagnosis (E; L); they are angry at their pain and feel it is unjust (D; E; L; N), and they grieve for what is lost, including a sense of self and financial independence (D; E; F; G; H; I; K; L; M; N; O; P).


*“I used to be extremely fit…so to have to go where I am now—I’m like a shell of a person really”*

*(Participant A)*



*“I used to do 25,000 steps a day…felt really refreshed, always up at the crack of dawn, out with the dog twice a day, the gym…my mood was good…we used to do quite a bit of walking like in the Yorkshire Dales…I’ve scaled that down now to little parks”*

*(Participant C)*



*“I used to be so active…gym everyday, bike and swim, and very fitness and health-orientated. I used to love to do the garden…I don’t even go outside now”*

*(Participant F)*



*“I used to really like walking over the fields with the dog but I’ve not done it in 4 years”*

*(Participant I)*



*“My job…I was constantly up and down ladders…swimming twice a week, I used to cycle at least a couple of hundred miles a week, I used to do two martial arts—tai boxing and JKD…it’s just changed my life to be honest. I was quite a healthy weight. My weight has now rocketed to nearly 20 stone”*

*(Participant M)*



*“I could do anything—if you spoke to my husband he’ll tell you he’s got a different wife now”*

*(Participant P)*


#### 3.3.1. Established Phases of the Decomposition Reaction

The established process of a decomposition reaction on an individual’s lifestyle and interactions was supported by evidence across ICF domains. This was dominated clearly by the ICF body function and structure domain, although this domain was influenced and interacted with the two other domains, including the activities and participation domain and environment domain. There is an extremely strong interlinking between the ICF domains that goes beyond the themes described. The participants’ behaviour can be seen as largely responsible for this. There is a strong theme of fear avoidance behaviour throughout the dataset. Participants expressed a lack of confidence in their physical ability to carry out work responsibilities and avoid action in order to preserve energy (D; H; I; J; P), planning actions around how may feel in the future (D; H; J). Participant I describes not bothering to go anywhere: *“getting ready to go out takes a lot of energy out of me”*. The decomposition interaction between the domains is considered below.

##### Describing the Reaction

The decomposition reaction includes a process where activities and interactions are restricted and limited (C; D; F; H; L; M; P). This was expressed in various ways, affecting individual’s activities and their vocation. The illness was described as becoming a bigger part of the individuals life (E); it meant life was lived more at home (A), that having a good day is not possible (L). Further, it identified that experiences across the ICF domains became a reason not do to things, and this was regarded as a life lost (F; G; H). Individual expressions often contextualised this. For instance, Participant F explains, “*it’s [fibromyalgia] taken my life away…I feel like a burden…helpless…that’s why I stay in bed and keep out of the way*”. Participant B identified that life had become about managing pain and living with pain, which changed how they viewed their job and prevents the individual from having a job (B; C; I; L; P) or engaging in meaningful activities or relationships (O).

The close association between symptoms was central to the decomposition reaction; for example, poor sleep leads to worse mental health and worse pain (C; D; E; H; I; J; K; M). A specific connection was identified with agitation (D; M; N; O), intrusive thoughts regarding pain prohibiting activities (E), anxiety reducing confidence (E;I;P), amotivation and a lack of a desire to live (F; K; P), low activity levels (G;P), low motivation to leave the house (G), paranoia and a lack of trust (K), and difficulties becoming motivated to perform basic tasks (M). One participant stated that they were very low, with depression and anxiety (A), and another identified how their mental health and their experience of pain were associated in a vicious cycle (D). Individuals could be low to the point of thinking about suicide. Participant P stated, “*he can’t cope with me saying things like I just can’t stand this anymore, I can’t live like this anymore, so it’s difficult for me to speak to him, do you know what I mean…he’d sit there and listen but I know it would hurt him and he’d worry all the time*”.

Individuals expressed difficulty in making plans due to the unpredictable nature of their symptoms. This included a loss of spontaneity (D; P), having no interest in people (F), expecting not to be able to make it through an event and pre-emptively thinking of excuses (G), events not being worth it due to after-effects (H), and social anxiety (I; K; P). Other resultant thoughts and behaviours included fear avoidance behaviours and fear of loss. Individuals also identified a high dependence on medical environments for basic exercises for reassurance, and for advice on therapies, even if complementary (E). Individuals recognised factors which influenced activities and interactions; this included (1) a need to plan for the pain that will come with physical movement (D; F; I; M; O; P), whilst acknowledging that it could be unpredictable (A; B), could be disproportionate to the activity choice (G), and could be affected by the weather (D; E; J; K); and (2) a need to use rest as a way to manage pain caused by movement (A; I; K; M) and to factor in rest days after social activities (I).

#### 3.3.2. The Experiences of Seeking Treatment and Change

Individuals had experiences of seeking treatment and attempting to make changes to their situation. However, individuals identified a low motivation and energy towards active therapies (H; L; M; N; O; P), being scared about active therapies (E; K; L; M; O), and mentally active coping strategies (as opposed to physical) (E; I; J; N). Some individuals also identified that they were not able engage in passive/distraction techniques (F); others identified that too much pain and exhaustion was experienced during physical activity and active therapies (F; K; L; M; O). The experiences of seeking therapies was also impacted by detrimental effects from relying on aids (G), relying on aids and carers from a young age (E; I), and passive strategies in the workplace (K; L; P). Two types of experiences were most often referenced: problematic experiences and ineffective experiences. This track record of negative experiences reinforces and pushes patients further into the negative cycle—they now become wedged in at the “bottle-neck”, and it becomes exponentially more difficult to escape.

##### Problematic Experiences

Problematic experiences represented medication-related, medical intervention-related, healthcare system-related, and healthcare professional/patient-related experiences. Medication-related experiences included a development of tolerance and subsequent dose escalation (C; D; E; I; M; N) to the point of taking far more than the recommended dosages (M); concerns with addictive potential (L); withdrawal (I; J; K); side effects (D; E; I; J; L; M), with a resultant loss of independence (D); a lack of benefit (K; L; M; N; O); a dislike associated with the need to be on medications (D; P); trying multitudes of different medications and not knowing which creates withdrawal effects from one versus the side effects of starting another (M); medication to help sleep but not experiencing quality sleep (D); medication makes things harder and impeding social activities (D;J); medication side effects interfering with clear-headedness and motivation (E), or creating slowed speech and impeding social interactions (J); and an inability to engage with other therapies secondary to medications—drugs inhibiting the ability to adopt self-management techniques (E). Intervention-related experiences included a lack of long-term benefit from medical interventions (D; I; J; K; L; M; O); seeking the same procedures despite no benefit, representing an element of desperation and no alternatives (L; O); requiring widespread injections for widespread pain (E; K); injections being painful (B); being wiped out for few days after both the effort of going for a treatment and due to the side effects (I); and feeling the need to rest for days after an intervention (M). Healthcare service-related experiences include the stress of waiting for referrals (A; O; P), and using NHS resources secondary to side effects (L). Healthcare professional/patient-related experiences include medical therapies not helping to manage other symptom domains of fibromyalgia (D; E; F; H; I; J; M; O), and patients (and clinicians) not being good at reflecting on other symptom domains when evaluating treatments (E).


*“I was on so much medication I didn’t know where I was. I didn’t know this at the time because I was taking all the medication”*

*(Participant A)*



*“The botox was the most painful thing I’ve ever had done in my life”*

*(Participant B)*



*“I’ve tried a lot of drugs…felt constantly hungover—that’s not something I want in my system for no reason…I don’t want more toxins in my body. I wouldn’t drive because they make me drowsy, I don’t want to lose my driving license”*

*(Participant D)*


##### Ineffective Experiences

Participants described a continual search for treatments, often despite trying all National Institute for Health and Care Excellence (NICE) recommended therapies (C); seeking warmth (D; K; L; M; P); trying so many gadgets they forget they are there (J); and individuals finding it difficult to understand the rationale for “education” on FM if it is not explained well by healthcare professionals (C). The search continued for individuals despite the recognition of potential harmful effects, including regular sunbeds (D) and internet-bought tablets (K). Both medical and alternative therapies were associated with a lack of efficacy: frequently, non-medical therapies were used in conjunction with medical therapies, yet still no effect was observed (D; I; L; O); there was a general recognition that most treatment experiences had short-term effects (E; H; J), and positive effects would become shorter in duration with each treatment (D). Some individuals identified that their symptoms seemed worse after non-medical therapies (I; J; L). There can be a lack of engagement with an active approach, and when NHS-based treatments were considered, some problems were reported. For instance, having space at home for at-home physiotherapy exercises (C); physiotherapy engagement tends to be reactive rather than proactive/preventative (C). There was a perceived lack of whole-body and holistic approaches: some identified shortcomings with complementary therapies, describing even the more holistic therapies to focus only on small anatomical areas and not the whole picture, including all the symptom domains (C). However, when a whole-body approach was employed, individuals identified that they required widespread treatments for widespread pain, e.g., lots of needles in acupuncture (B; I), which in itself could be problematic. Participant E identified that they would seek medical over non-medical interventions. One reason for this was being scared of acupuncture but not widespread medicated injections with larger needles, *“but I think I see it differently ‘cause it’s not medical”.* Barriers to treatments were identified: not being presented alternative therapies as a treatment option (O), and a difficulty identified by not being to able receive many, if any, alternative therapies through the NHS, as without this, they would be too expensive (I; K; L).

Two individuals identified some value in treatments. Participant D identified the value of lidocaine infusion: *“so before I’d feel very hunched and stiff, whereas if I have the infusion I kinda feel a bit lighter and less sluggish and I seem to have less headaches”*. Participant E identified the value of a pain management course and from meditation. Finally, Participant H identified value in hydrotherapy and cupping, which benefits their energy levels.

### 3.4. Potential for Recomposition Reaction

The identification of ICF domains that enabled a recomposition reaction and allowed activities and interactions to be engaged in was less well presented. The two domains considered were activities and participation, and the environment domain.

#### 3.4.1. Describing the Reaction

The dataset revealed a small amount of evidence to show that participants held a potential to find their way out of the downward spiral. Represented in Figure 2, a minority of individuals showed promise with regard to finding methods to enable their activities, and finding a support network. However, no evidence was expressed for the theme of “body structure and function”, and hence, this item remains “trapped” at the bottle-neck of the cycle, whilst the other two domains are seen to be finding their way out of the cycle, albeit on a small scale.

#### 3.4.2. Activities and Participation

1. Adjustments made to enable activities: One individual focused on how it was possible to adapt to the challenges faced. Participant N explains, “*adjustments are needed to make life work, a scooter to go out, a walk-in shower, comfort crutches for walking*”. Participant C identifies the benefits of being active on their mental well-being, stating, “*you’ve gotta do best by the dog so that keeps me active. Sometimes on the days when I’m really bad and I can’t, [name] will take me out in the car and I’ll sit on the bench and he’ll walk the dog around, but I am still out kinda thing. So I am getting there*”. Another individual identified doing something, even if small, which was valued and enjoyed. Participant N states, *“I’ve got into gardening … I do my little bit and I do enjoy that”.* Participant P identifies walking as an adapted activity, and states, ”*so like last Wednesday I try and walk from my house to my sister’s…I’ve got my grandson’s buggy so I hold on to that…so I try and do that walk every Wednesday but it ain’t easy. But I do it because I wanna take my grandson for a walk—he shouldn’t not have fresh air because I can’t do it”, “when I seen the doctor last time he said to me you need to use a walking aid…but it’s just stubbornness*”.

2. Vocational achievements: Participant E focused on what had been achieved as a source of comfort, explaining, “But I got my GCSEs [General Certificate of Secondary Education]…I went to 6th form, I went to university, got my degree—Psychology and English”. Participant H identified the value in gardening and the results of that: “well my gardening, that’s what helped me with the fibro, you know seeing like my potatoes, my corn, there’s tomatoes and runner beans so like when I reap that it makes me feel good you know, happy…fulfilled”. Alternatively, Participant O identified her actions towards others which had value, stating, “I try and read or I knit if my hands will let me. I knit hats for the special care baby unit at Birmingham. The little premature hats”.

3. Activity-related achievements: In a similar way to vocational achievements, activities that were enjoyed appeared to help energy. Participant J stated, “*sometimes I think to myself I really enjoyed that, that’s given me a bit of energy or that’s made me feel better about the day*”.

#### 3.4.3. Environment

Finding value and strength in relationships: Individuals highlighted the value of relationships with others (A; J). For instance, Participants A and K describe the benefits of seeing their grandsons—”that gives me a high” or “keeps me going”. Further to this, Participant M highlighted the value of someone else understanding, stating, “*I mean my wife has got fibro as well so we sort of understand one another…I think that’s the saving grace to be honest because we’ve got respect for one another*”.

## 4. Discussion

It is our belief that the present study is the first to attempt to bring all of the FM symptom domains together into one model: a model that not only highlights the processes and complex interactions of FM lived experiences, but a pragmatic model to aid in bringing about much-needed change and updates in the management of our FM patients into clinical practice.

### 4.1. Cognition to Feature in the FM “Vicious Cycle”

The present study highlights a close association and interdependence between all facets of all symptom domains. FM is seen to affect participants in every way—physically, mentally, socially, and cognitively, to name a few. Low moods dictate activity levels, energy, and motivation. Low moods are seen to develop secondary to physical inability, poor sleep, and subsequent fatigue, which further exacerbates the negative feedback loop, and so on and so forth. Ultimately, reduced confidence in physical and social abilities are inextricably linked with all symptom domains—it is a major player in the entry to and descent through the negative and vicious cycle—which then dictates participants’ behaviours and fear avoidance, driving them yet further down. The chronic pain cycle has long been reported and utilised in patient manuals, namely “The Pain Toolkit” [54], as well as biopsychosocial factors perpetuating chronic pain [55]. For example, sleep and depression are commonly identified as impacting FM symptoms, and vice versa [56,57]. A novel finding from our data is the significant and often debilitating impact of cognitive problems, which is seen to exacerbate the “vicious cycle” at all levels.

### 4.2. The Fibromyalgia Decomposition Phenomenon

Here, we present a novel phenomenon. This work has undergone a thematic analysis to bring out a range of themes and sub-themes. It is unique in that we have taken the analysis several steps further, bringing out a common process that FM patients endure—“the Fibromyalgia Decomposition Phenomenon”. We have brought together multiple considerations as a process which impacts on choices and one’s ability to be active. We delved even deeper into this complex process, and begin to bring out the data in a way that can be utilised towards future treatments and looking for “recomposition” strategies from an early stage. We have gone on to begin to briefly describe potential sub-themes for this according to the ICF framework.

Decomposition is traditionally a scientific term relating to biological, chemical, and physical processes. The exact definition varies according to context, but the process generally involves the breakdown of a compound or reactant into multiple products [58]. It entails a complex set of different processes “acting upon a wide variety of organic substrates that are themselves constantly changing” [59]. Even computers cannot begin to solve problems without a decomposition process: complex problems need to be broken down into more manageable parts before understanding can occur [60]. Decomposition involves an immense diversity of possible factors and interactions. It occurs simultaneously but at different rates—some components disappearing rapidly, some slowly, and some only after a time delay. It is a far more complex and heterogeneous process than simply the degradation or decay of, for example, a radioactive isotope [59]. This terminology has not been used before to describe the breakdown of a person as a whole into smaller segments that no longer appear to work together in unison. The common catalyst in these cases seems to be the diagnosis of FM. Just as biodiversity significantly alters the decomposition process [61], here, the patient is the organic substrate which is constantly changing with the condition, with the agents acting on it being external factors, environment, relationships, and so on. Patients are seen to enter this decomposition cycle readily. It is disappointing to see that no potential “way out” has been identified to improve body structure and function, despite a wide range and number of treatments trialled in this population. The “lost cause”, our participant demographic experience, demonstrates a succumbing to the diagnosis, with engagement in physical activity being deemed out of their control; this is perhaps a reflection of the severity of FM in our participant demographic, where the median FM severity scores were towards the high end of severe.

### 4.3. Potential for “Recomposition”

The definitions of recomposition include the recombination of constituents, to reorganise, rearrange, or reform, and to restore composure or calmness [62]. Body recomposition is described as a weight loss technique by some, where the aim is to tip the balance of adipose tissue/muscle tissue [63]. Our definition of recomposition goes beyond the physical sense. We are referring to individuals with FM learning new ways of acting. There needs to be an acceptance that some parts of them are lost, so they should not be the exact image of their “old self”, but the majority of constituents still exist, waiting to be remodelled and recycled, and to learn new ways of acting. As with decomposition, some nutrients and energy are imported to synthesise new substances [59]. This decomposition/recomposition cycle is almost akin to the grief cycle. If patients and clinicians have an awareness of its existence, it has the potential to guide FM patients through the motions quicker and reach the recomposition process at an earlier stage, or even to reach the process at all. The grief cycle has previously been utilised in the preparation of organisational change in order to plan a change strategy for it to run smoothly and optimally [64].

Many participants feel as though the future is uncertain and unknown, and hence, cannot see a way out from the bottom of the decomposition spiral. Preconceived ideas towards a course of whole-body PBMT and, potentially, a way out of the spiral, included (1) all other treatments failing, and so looking for anything else (K); (2) hope at the possibility of a new treatment (A; B; C; I; K; L; M; P); (3) a treatment being better for, and easier on, the body than medication (D;L); (4) looking for any link in the chain to be dealt with in order to have a knock-on effect on rest of one’s symptoms (M); and (5) more energy and hence less pain, and therefore an ability to return to being one’s old self—*“I just wanna go back to being me. I just wanna be me. And at the moment I’m not me” (Participant P).* Our data show that physical ailments are certainly a substantial barrier in this more severe cohort of FM patients—we need to find ways to overcome these if recomposition can be successful.

### 4.4. Requirement for a Widespread and Holistic Approach Not Only to Treatment, but in Patient and Clinician Treatment Evaluation

Participants describe their symptoms as being widespread and affecting their whole body, and express concerns that only small sections of their body are targeted at any one time—*“it’s horrible pain, you can’t describe it—it’s the worst thing I’ve ever had…nerve pain, sensitive skin, tingling, and I jump as well…even in me toes, me toes tingle”; “my whole body just felt very exhausted, it became like a heavy weight”; “I feel like my whole body is hot”.* Emerging evidence supports this notion and describes widespread small-fibre neuropathies in FM [65] that are difficult to target any one treatment towards. Furthermore, it has long been known that neuropathic pain is complex, difficult to treat, and can become refractory, requiring a multi-mechanistic pharmacological treatment approach [66]. Future treatments should focus on the whole body as one entity in order to treat all facets highlighted in Figure 1. Interestingly, participants, “word for word”, expressed this desire for a more integrated approach. Dealing with FM promptly and as one entity has the potential to reduce patient stress, improve treatment efficacy, and, possibly, prevent patients sinking deeper into the decomposition spiral, which will ultimately save healthcare services time and money. There is an urgent need for a more integrated approach.

Additionally, our data highlight that even well-educated patients with good health literacy are not good at reflecting on any other symptom domain than pain, even when prompted. This has the potential to affect evaluations of new treatments, as the interruption of any part of the cycle can indirectly reduce pain [67] and, therefore, should not be overlooked by patients and healthcare professionals alike.

### 4.5. Limitations

Pain catastrophising was not controlled for in this study, yet it was clear from the interviews and B.C.F.’s clinical experience which participants would have scored highly on the Pain Catastrophizing Scale. The weight of this implication is likely to surface in the mid- and post-treatment interview analysis—it is likely to have bearing on which participants have a lower threshold for cycling back round to the decomposition cycle. Past research suggests that high catastrophising scores correlate with activity engagement, and that subjective assessments can be poor indicators of physical functioning in these individuals [68].

### 4.6. Recommendations for Future Research and Clinical Implications

A common feature arising was participants’ appreciation and relief to be diagnosed and therefore validated, often after a long journey of anguish and consultation with multiple specialties. Anecdotally, staff lack confidence in giving the diagnosis, and there is a common misconception that the final step in the pathway needs to come from a rheumatologist. This is validated in recent Royal College of Physicians guidance, which advocates for early diagnosis, but suggests that clinicians can feel overwhelmed [2]. In actual fact, NHS England now advocates for community care for non-inflammatory painful conditions, including FM [69]. This supports the theory and practice of the newly instituted integrated care systems [45], in order for patients to receive integrated care and services.

There is a need for more timely diagnosis in FM patients [2,44], albeit with a recognition that a diagnosis alone is insufficient to capture what is important to patients [24]. The Faculty of Pain Medicine acknowledges the complexity of new patient consultations, recommending that new patient consultations should take one hour [70], recognising the need to assess the medical, psychological, and social influences on the experience of pain, and to consider the patient’s individualised needs and perspectives [71]. Yet healthcare professionals continue to feel that there is a lack of time allocated in order to give quality consultations [44]. Extending initial consultation times and offering a follow-up appointment to help patients come to terms with, and to guide them through, the decomposition process that has occurred/is occurring could allow patients to reach the recomposition process quicker, and therefore, save on future multiple repeated consultations, “treatment-shopping”, and dissatisfaction down the line.

There is a gap in how we classify function and disease-related disability. The concept of disability is complex and interpreted in a variety of different ways [72], and is not so “black and white” as simply whether a patient is in receipt or not of disability benefits. The updated ICD-11 came into play in 2019. Prior to this, classification and coding of chronic pain was poor [73]. To complicate matters, the ICF is intended to complement the ICD [74]. Anecdotally, there is a lag in the uptake of the new ICD, as well as a lack of awareness of the ICF. The ICF is meant to create a broader and more meaningful picture of health experience—something which FM lends itself to. The mere act of completing the ICD combined with the ICF has the potential to improve patients’ consultation satisfaction, and is a quick and simple way for the treating clinician to glean important information about their patient’s functional limitations to help target treatments, as well as serving as a reference system at the population level. As such, the ICF addresses patient concerns with regard to stigmatisation, as broad, recognisable, and patient-centred language is used [74]. Despite this, the ICF is seldom used, either clinically or in the literature. It fosters a holistic view of a person’s lived experience—its use has the potential to address concerns raised both in this dataset and in the PACFiND study [44], as well as facilitating appropriate treatment. As aforementioned, the PACFiND study [44] identified a lack of services and referral pathways for FM. Simply improving coding to represent the true gravity of the “FM epidemic” could highlight the substantial unmet need to policy makers. The Revised Fibromyalgia Impact Questionnaire could also act as a surrogate marker of functioning associated with FM symptoms, giving an overall figure, which would be easily comparable at future consultations.

Our data show that patients are dissatisfied with not being treated as a whole; they continue to seek more and more treatments, which adds to the overall psychological burden.

A recent large study revealed a preference towards non-pharmacological therapies for the management of FM symptoms [75]; however, our data and the existing literature show that both efficacy and accessibility remain a barrier [76]. We need to look for more cost-effective treatments that target the whole picture. We know that there is a lack of multidisciplinary holistic services in the UK [44]—whole-body PBMT is a simple way of addressing this [44]. In future work, relating to treatment with whole-body PBMT in this patient group, we hope to capture a more detailed understanding of the recomposition cycle.

## 5. Conclusions

From this small participant cohort, we conclude that healthcare practitioners should acknowledge the decomposition/recomposition cycle following diagnosis. Simply providing the diagnosis alone is not adequate. Patients should be guided through the cycle in the hope of a quicker entry into the recomposition phase. Utilising ICD-11 in combination with the ICF will help both the patient and clinician understand the functional impairment at that time, and compare it with future assessments. We have long known that FM is not as simple as addressing pain levels, but patients and clinicians alike continue to use this as a basis for their management. We postulate that shortening the timeline in the FM patient pathway holds potential to reduce disease-related co-morbidity and functional impairment, that is, both an earlier diagnosis and a broader, more holistic treatment approach.

## Figures and Tables

**Figure 1 behavsci-14-00047-f001:**
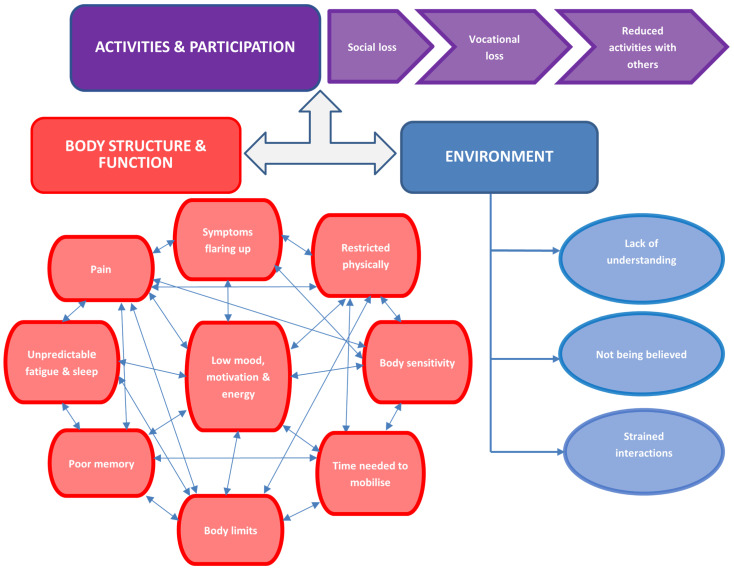
Themes and sub-themes derived from study’s dataset.

**Figure 2 behavsci-14-00047-f002:**
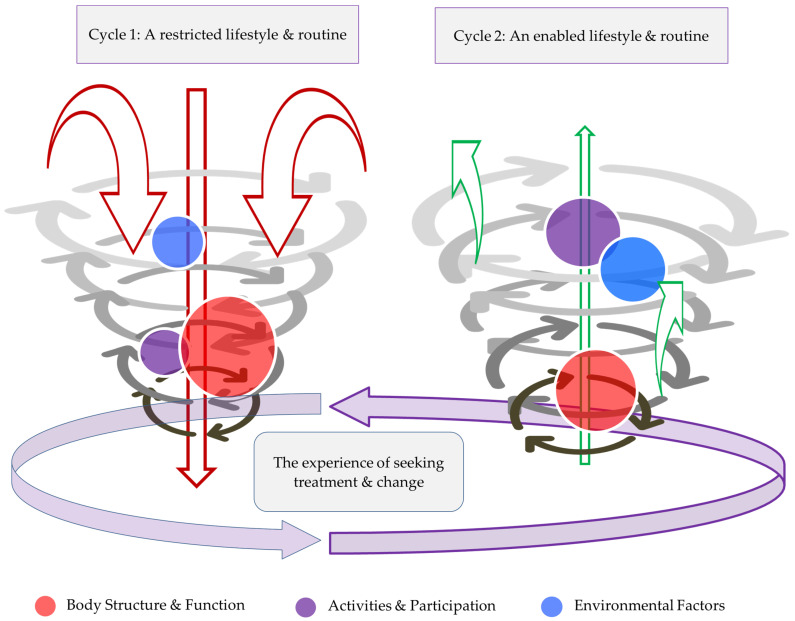
The decomposition phenomenon, with potential for recomposition. The model illustrates the ICF domains (themes) and sub-themes interacting to cause a decomposition reaction within an individual’s lifestyle. Cycle 1 represents participants exhibiting a definite and prominent entry into the downward spiral from onset of their symptoms and diagnosis. Each theme is represented by a coloured circle of varying sizes to reflect its intrusiveness. Cycle 2 shows the potential to emerge from the negative spiral. It should be noted that cycle 1 compared with cycle 2 is not to scale. In fact, when contrasting proportion of decomposition/recomposition evidence within our dataset, recomposition accounts for only 2% of the data. Of the horizontal arrows to show connection between the two cycles, the arrow on the righthand side is more prominent to reflect the very low threshold for participants to return back to the negative cycle when, for example, a new treatment has not “worked”.

**Table 1 behavsci-14-00047-t001:** Participant demographics.

Participant ID	Gender	Age (Years)	Symptoms (Years)	Marital Status	Employment	Education Level	Ethnicity	BMI Category (kg/m^2^)
A	Female	51–55	27	Married	Unemployed (looking for work)	Further education (sixth form)	White British	30.0 to <35
B	Male	46–50	6	Single	Self-employed	Higher education	Asian/Asian British	18.5 to <25
C	Male	41–45	6	Civil Partnership	Self-employed	Higher education	White British	25.0 to <30
D	Female	31–35	10	Single	Sick leave	Higher education	White British	18.5 to <25
E	Female	36–40	26	Single	Unemployed (not looking for work)	Higher education	White British	25.0 to <30
F	Female	51–55	10	Married	Employed full-time	Higher education	White British	35.0 to <40
G	Female	31–35	24	Single	Unemployed (not looking for work)	Higher education	White British	35.0 to <40
H	Female	46–50	10	Single	Unemployed (not looking for work)	Higher education	Black British-Caribbean	>40
I	Female	26–30	4	Co-habiting	Unemployed (not looking for work)	Higher education	White British	25.0 to <30
J	Female	61–65	31	Divorced	Retired	Higher education	White British	25.0 to <30
K	Female	51–55	17	Married	Employed full-time	Secondary school	White British	25.0 to <30
L	Male	36–40	24	Married	Employed full-time	Higher education	Asian/Asian British	25.0 to <30
M	Male	46–50	11	Married	Retired	Higher education	White British	>40
N	Male	66–70	15	Married	Retired	Secondary school	White British	35.0 to <40
O	Female	66–70	15	Married	Retired	Some secondary school	White British	35.0 to <40
P	Female	51–55	10	Married	Employed full-time	Higher education	White British	35.0 to <40

## Data Availability

The datasets used and/or analysed during the current study are available from the corresponding author on reasonable request.

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
