# Peer review of "The Fibromyalgia Decomposition Phenomenon: A Reflexive Thematic Analysis"

_behavsci, 2024, doi:10.3390/bs14010047_

Round 1

Reviewer 1 Report

Comments and Suggestions for Authors

Dear authors, it’s an interesting article but can you please re-write the abstract and make it more clear what it is about and also you need to use the limitation of your study. I.e. it is a very small study and that needs to be reflected in your conclusion and in the conclusion in the abstract as well.

Author Response

Response 1: it’s an interesting article but can you please re-write the abstract and make it more clear what it is about

Comments 1: We have added a sentence to more clearly explain the aim of the study. 

Response 2:  also you need to use the limitation of your study. I.e. it is a very small study and that needs to be reflected in your conclusion and in the conclusion in the abstract as well.

Comments 2: Thank you for this. We have addressed this in both the abstract and main text conclusion. 

Reviewer 2 Report

Comments and Suggestions for Authors

Dear Author's

I read the article with great interest. However, I see some aspects of it that, in my opinion, require additions / improvement namely:

1. In the abstract, please add 1-2 sentences, any conclusions from the research ... currently none

2. 16 participants ... this is not representative, it should be included in limitations ...

3. references - many items, although some are over 20 years old - history ...

4. The article is very long ... (possibly shortened), such long texts are tedious for a potential reader ... (to think about) ...

Best regards

Author Response

Comment 1. In the abstract, please add 1-2 sentences, any conclusions from the research ... currently none

Response 1. Thank you. We have added 2 sentences briefly explaining the implications. 

Comment 2. 16 participants ... this is not representative, it should be included in limitations ...

Response 2. Thank you, we have now included this both in abstract and main text. 

Comment 3. references - many items, although some are over 20 years old - history ...

Response 3. The older of the references are describing i) development of a questionnaire. ii) highlighting an old definition that has since been updated. 

Comment 4. The article is very long ... (possibly shortened), such long texts are tedious for a potential reader ... (to think about) ...

Response 4. Thank you. We have added a sentence at the beginning of the results explaining the length and directing to the figures and model as a focus for implications. 

Reviewer 3 Report

Comments and Suggestions for Authors

This is a topical and important issue: fibromyalgia is a painful and chronic condition with delayed diagnosis and treatment, which affects quality of life and biopsychosocial aspects. We are interested in knowing the perceptions of people with fibromyalgia, but above all we are interested in knowing how they cope and in proposing person-centred interventions to support the management of their daily situation.

The title is succinct, indicative of the content of the study, although I would change it to 'recomposition' in a positive way.

The abstract allows to quickly identify the basic content, but in a very general way, without highlighting the specificities of this study, clearly describing the objective, the methodology, the results, although the reference to decomposition or recomposition is missing. Eliminate the acronym (FM).

The review conducted is relevant. Keep the review organised and avoid self-citations.

The structure of the article is not logical, I think it is convenient to explain the phenomenon of fibromyalgia decomposition in the introduction to understand the reason for the research and not have to wait for the discussion. The aim and the discussion should be more closely related. I see this as a study that is part of a larger research related to the PBMT intervention.

Acronyms should be explained the first time they are used (e.g. EULAR, UK, NHS, NICE, GCSE).

The wording of the manuscript should focus more on the recomposition reaction potential (3.4) and less on the problems.

The methodology is appropriate and in line with the theoretical rationale. The data collection is adequate.

The data is analysed in relation to the aim of the study. The graphs and tables clarify the information. More emphasis should be placed on recomposition.

Findings sections should be discussed in relation to the aim, or the aim should be worded more in line with the discussion.

References are appropriate and up to date.

Eliminate self-quotes. References 13 and 16 are the same document. Reference 20 is missing the year (2010)

Author Response

Comment 1: The title is succinct, indicative of the content of the study, although I would change it to 'recomposition' in a positive way.

Response 1: Thank you for this. It is thought-provoking, however, I think the baseline study is mostly focusing on the decomposition process with minimal evidence for recomposition at this point. We hope to talk more positively on recomposition on our follow up paper post-intervention. 

Comments 2. The abstract allows to quickly identify the basic content, but in a very general way, without highlighting the specificities of this study, clearly describing the objective, the methodology, the results, although the reference to decomposition or recomposition is missing. Eliminate the acronym (FM).

Response 2. We have eliminated the acronym FM in the abstract. We have added in an aim in the abstract as well as more definition conclusion mentioning decomposition:recomposition. 

Comments 3. The structure of the article is not logical, I think it is convenient to explain the phenomenon of fibromyalgia decomposition in the introduction to understand the reason for the research and not have to wait for the discussion. The aim and the discussion should be more closely related. I see this as a study that is part of a larger research related to the PBMT intervention.

Response 3. Thank you. We have now directed the reader more easily to relevant points in the results. We feel it is difficult to talk about decomposition in the introduction as it is a new phenomenon derived secondary to the results of this study. Correct - this is a baseline qualitative study and the follow up qualitative results post-intervention are being analysed. 

Comment 4. Acronyms should be explained the first time they are used (e.g. EULAR, UK, NHS, NICE, GCSE).

Response 4. All acronyms have now been explained. 

Comment 5. Findings sections should be discussed in relation to the aim, or the aim should be worded more in line with the discussion. 

Response 5. Thank you - we have reported in line with The Standards for Reporting Qualitative Research. 

Comments 6. Eliminate self-quotes. References 13 and 16 are the same document. Reference 20 is missing the year (2010). 

Response 6. Thank you for this, we have added in 2010. References 13 and 16 are slightly different and from different years. We have left the one self quote in as it describes the basis of the therapy in more detail (which is not explained anywhere else in this context). However, we can remove it if you do feel strongly about this. 

Round 2

Reviewer 3 Report

Comments and Suggestions for Authors

The summary has improved. The aim is now clearer. In the results you now refer to Figure 1 and Figure 2 in the first paragraph on page 6, but the figures are on pages 8 and 17. The figures should appear in the text close to where they are first mentioned or a short explanation of the two figures should be given without mentioning them. The self-citations 48 and 68 should be deleted.

Author Response

Comments 1: In the results you now refer to Figure 1 and Figure 2 in the first paragraph on page 6, but the figures are on pages 8 and 17. The figures should appear in the text close to where they are first mentioned or a short explanation of the two figures should be given without mentioning them

Response 1: Thank you. This sentence has been amended, and reference to figures removed at this point. 

Comments 2: The self-citations 48 and 68 should be deleted.

Response 2: Self-citations 48 and 68 are now deleted.